# Nutritional Status According to the GLIM Criteria in Patients with Chronic Heart Failure: Association with Prognosis

**DOI:** 10.3390/nu14112244

**Published:** 2022-05-27

**Authors:** Clara Joaquín, Nuria Alonso, Josep Lupón, Paloma Gastelurrutia, Alejandra Pérez-Monstesdeoca, Mar Domingo, Elisabet Zamora, Guillem Socias, Analía Ramos, Antoni Bayes-Genis, Manel Puig-Domingo

**Affiliations:** 1Endocrinology and Nutrition Department, Hospital Universitari Germans Trias i Pujol, 08916 Badalona, Spain; nalonso32416@yahoo.es (N.A.); alec148@gmail.com (A.P.-M.); guillemsociesr@gmail.com (G.S.); aeramos.germanstrias@gencat.cat (A.R.); mpuigd@igtp.cat (M.P.-D.); 2Department of Medicine, Universitat Autònoma de Barcelona, 08023 Barcelona, Spain; jlupon.germanstrias@gencat.cat (J.L.); ezamora.germanstrias@gencat.cat (E.Z.); abayesgenis@gmail.com (A.B.-G.); 3ICREC Research Program, Health Sciences Research Institute Germans Trias i Pujol, Universitat Autònoma de Barcelona, 08916 Barcelona, Spain; pgastelurrutia@gmail.com; 4CIBERCV, Instituto de Salud Carlos III, 28029 Madrid, Spain; 5Heart Failure Unit, Cardiology Department, Hospital Universitari Germans Trias i Pujol, 08916 Badalona, Spain; mdomingo.germanstrias@gencat.cat

**Keywords:** MNA, GLIM criteria, heart failure, prognosis, CV mortality

## Abstract

Background: The Global Leadership Initiative on Malnutrition (GLIM) criteria were recently proposed to build a global consensus on the diagnostic criteria for malnutrition. This study aimed to evaluate the GLIM criteria for its prognostic significance in outpatients with heart failure (HF), and to compare them to a previous validated method, such as the Mini Nutritional Assessment (MNA). Methods: This was a post hoc observational analysis of a prospectively recruited cohort, which included 151 subjects that attended an outpatient HF clinic. At baseline, all patients completed the nutritional screening MNA short form and the nutritional assessment MNA. In a post hoc analysis, we evaluated the GLIM criteria at baseline. The outcomes were based on data from a five-year follow-up. The primary endpoint was all-cause mortality. Secondary endpoints were cardiovascular (CV) mortality and recurrent HF-related hospitalizations. We also investigated whether the GLIM criteria had better prognostic power than the MNA. Results: Abnormal nutritional status was identified in 19.8% of the patients with the GLIM criteria and in 25.1% with the MNA. In the multivariate analyses (age, sex, NYHA functional class, diabetes, and Barthel index), nutritional status assessed by the MNA, but not by the GLIM criteria, was an independent predictor of all-cause mortality, CV mortality, and recurrent HF-related hospitalizations during the five-year follow-up. Conclusions: Malnutrition assessed by MNA, but not by the GLIM criteria, was an independent predictor of all-cause mortality, CV mortality, and recurrent HF-related hospitalization in our cohort of outpatients with HF.

## 1. Introduction

The reported prevalence of malnutrition in patients with heart failure (HF) varies widely among different studies (25–60%) depending on the method of nutritional assessment used, whether the patients were hospitalized, and the functional class of HF studied [1,2,3]. Regardless of the method of nutritional assessment applied, many studies showed that malnutrition was an independent predictor of mortality [4,5,6]. In order to standardize the diagnosis of malnutrition, the Global Leadership Initiative on Malnutrition (GLIM) recently developed and reported new universal criteria to diagnose malnutrition in adults based on a two-step model for risk screening and diagnostic assessment [7]. For patients considered to be at risk of malnutrition according to nutritional screening, the GLIM criteria recommend performing a nutritional assessment that includes phenotypic information (unintentional weight loss, low body mass index (BMI), and reduced muscle mass) as well as etiologic criteria (reduced food intake or assimilation, and disease burden). To diagnose malnutrition, at least one phenotypic criterion and one etiologic criterion should be present. In addition, the GLIM criteria also grade the severity of malnutrition as stage 1 (moderate malnutrition) or stage 2 (severe malnutrition). Because the GLIM criteria consensus has recently been proposed, it is important to validate them in various populations, including patients with HF, and to determine their capacity to predict adverse clinical outcomes. Recently, malnutrition defined according to the GLIM criteria was reported to be a predictor of both low physical function and mortality in hospitalized patients with cardiovascular disease [8]. To our knowledge, there are no studies that evaluate the GLIM criteria in relation to prognostic significance in HF outpatients.

In a previous pilot study, we compared two established and validated nutritional assessment tools (Mini Nutritional Assessment [MNA] vs. Subjective Global Assessment [SGA]) in terms of prognostic significance in ambulatory subjects with HF. In that study, we observed that malnutrition assessed with the MNA, but not the SGA, was an independent predictor of mortality at two years in outpatients with HF, and also had implications in QoL and physical disability [9].

The purpose of the current study was to assess the prognostic significance and predictive capacity of malnutrition identified with the GLIM criteria for overall mortality, cardiovascular mortality, and recurrent HF-related hospitalizations in outpatients with HF. We also evaluated whether the GLIM criteria had better prognostic power than the MNA.

## 2. Methods

### 2.1. Study Design

This is a post hoc analysis from an observational prospective cohort study comprising a sample of 151 subjects that attended an outpatient HF clinic at a university hospital from June to December 2013. In a previous study, a sample size of 149 subjects was calculated to detect a statistical difference over 13% in the prevalence of malnutrition or risk of malnutrition between MNA and SGA in a cohort of HF patients, estimating a proportion of 20%. Furthermore, an alpha risk of 0.05 and a beta risk of 0.2 in two-sided tests were accepted.

All patients had an established diagnosis of HF, according to the European Society of Cardiology guidelines [10]. The criteria for a referral from clinical practice to the HF unit have been described previously [11,12]. We excluded patients with cancer, acquired immune deficiency syndrome (AIDS), liver cirrhosis, and chronic renal failure that required hemodialysis (see study flowchart in Appendix A). Demographic and clinical data were prospectively collected at enrollment. Patients were followed for 5 years.

The primary endpoint was all-cause mortality at 5 years. Secondary endpoints were the number of recurrent HF-related hospitalizations and cardiovascular-related mortality. Death was considered to be cardiovascular-related when it was caused by any of the following: HF (decompensated HF or treatment-resistant HF, in the absence of another cause); sudden death (unexpected death, witnessed or not, of a previously stable patient with no evidence of worsening HF or any other cause of death); acute myocardial infarction (death directly related to acute myocardial infarction, whether due to mechanic, hemodynamic, or arrhythmic complications); stroke (associated with recent acute neurologic deficit); procedure (post-diagnostic or post-therapeutic death); or other cardiovascular causes (e.g., rupture of an aneurysm, peripheral ischemia, or aortic dissection). Fatal events among patients with HF were identified from records maintained by the clinic, hospital wards, the emergency room, general practitioners, or by contacting the patient’s relatives. Furthermore, data were verified by comparing them to information in the Catalan and Spanish Health Systems’ databases. Adjudication of events was performed, individually, by two of the authors. This study was approved by the local Ethics Committee, and informed consent was obtained from each patient at enrolment for future general research regarding HF. The investigation conformed to the principles outlined in the Declaration of Helsinki.

### 2.2. Nutritional Screening and Assessment

Nutritional status was screened at baseline with the Spanish version of the MNA short form (MNA-SF) [13], and also evaluated in all subjects by MNA questionnaire [14]. The MNA form was provided by Société des Produits Nestlé SA 1994, Revision 2009, Vevy, Switzerland, Trademark Owners, which holds the copyright (http://www.mna-elderly.com/, accessed on 18 November 2021). The data used to complete the tests were obtained by a physical exam and a personal interview conducted by a trained nutritionist. In this post hoc analysis, we evaluated the GLIM criteria on data collected at baseline. For the first stage of these criteria, which involve nutritional screening, we used the MNA-SF. The second step involved diagnosis of malnutrition based on three phenotypic and two etiological components. The phenotypic components are: (i) non-volitional weight loss >5% within the past 6 months or >10% beyond 6 months; (ii) BMI <18.5 kg/m^2^ for age <70 years or <20 kg/m^2^ for age ≥70 years; and (iii) reduced muscle mass. The GLIM criteria suggest that a reduced muscle mass should be diagnosed based on a body composition assessment (e.g., bioelectrical impedance analysis, computed tomography, or dual-energy X-ray absorptiometry). However, standard anthropometric measurements, such as arm muscle circumference (AMC), are also acceptable in the clinical setting. To assess reduced muscle mass, we used arm muscle circumference (AMC) and hand grip strength as a supportive measure. AMC was calculated by the following formula: AMC (cm) = arm circumference (cm) − [0.314xtriceps skinfold (mm)]. Measurement of triceps skinfold was performed in triplicate by the same investigator using a Holtain constant pressure caliper (Holtain Limited, Crymych, UK) on the dominant arm. The mean of the three measures was calculated for the analyses. We considered that a patient had a reduced muscle mass when the AMC value was lower than the fifth percentile (p5) for each age and sex group of the reference population [15]. Hand grip strength was measured three times bilaterally with the Jamar^®^ dynamometer (in the second handle position). For this test, the individual was positioned with the shoulder adducted to zero degrees of rotation, the elbow flexed to 90 degrees, and the wrist in the neutral position. The average of the three measurements was used for the analyses [16]. Values under the fifth percentile of the Spanish normative reference data were considered as reduced muscle strength [17]. The etiological components of the GLIM criteria are: (iv) reduced food intake (<50% of energy requirements during >1 week or any reduction for >2 weeks) or assimilation and (v) inflammatory status. Reduced food intake was evaluated at baseline using quartiles by a dietitian and food assimilation per clinical record. Chronic disease-related inflammation was identified as a C Reactive Protein (CRP) concentration >5 mg/L.

Malnutrition was diagnosed based on the presence of at least one phenotypic criterion combined with at least one etiologic criterion.

Based on previous experience [9], we used MNA as the gold standard against which to compare the GLIM criteria. Those patients with moderate or severe malnutrition by GLIM criteria or those with malnutrition or risk of malnutrition by MNA were merged and considered as having abnormal nutritional status for statistical analysis.

When a malnourished patient was identified, the dietitian assessing nutritional status took action, and tailored recommendations were given to the patient based on the standard-of-care criteria.

### 2.3. Other Clinical and Analytical Parameters

Patient HF clinical status was based on the New York Heart Association (NYHA) functional class and the duration of HF.

To assess physical disability, we used the Barthel index [18], a standardized scale that evaluates the degree of dependence on assistance for performing the basic activities of daily living (range 0–100).

For a comprehensive assessment of nutritional status, biological biomarkers (CRP, cholesterol, albumin, and total lymphocyte count) collected at inclusion were included in the study. Blood samples were collected by venepuncture, between 8:00 and 08:30 AM, after an overnight fast.

### 2.4. Statistical Analysis

Categorical variables are expressed as frequencies and percentages. Continuous variables are expressed as the mean ± standard deviation or the median and 25th–75th percentiles. The normality of a distribution was assessed with the normal Q–Q plot. Statistical differences between groups were assessed with the chi-square and Fischer’s exact tests, for categorical variables, and with Student’s *t* tests, for continuous variables. To assess the prognostic significance of the nutritional assessment tools, we created two separate multivariable Cox regression models (enter method), with all-cause mortality as the dependent variable and the MNA or GLIM criteria score as the independent variable. Variables that showed statistical significance in the univariate analyses were taken as independent covariates. We also included several clinically relevant variables as covariates (age, gender, NYHA functional class, Barthel index, and Diabetes Mellitus). Cox survival curves were plotted to ascertain the relationship between the baseline presence of malnutrition and mortality. In all analyses involving cardiovascular-related death, competing risk strategy using the Gray method was adopted, and non-cardiovascular events were considered the competing events for cardiovascular-related death.

Recurrent HF-related hospitalizations are expressed as crude incidence rates (i.e., the number of hospitalizations per 100 patient-years). A binomial negative regression (univariable and multivariable) was performed, and results are expressed as the incidence rate ratios (IRR) and 95% confidence interval (95%CI). For the later analyses, out-of-hospital death due to HF was considered as an additional event.

To evaluate GLIM criteria, we used the MNA as the gold standard following our previous research [9]. We compared both nutritional assessments tools by evaluating their agreement and Cohen’s kappa correlation index.

Statistical analyses were performed with IBM SPSS Statistics 24.0 (IBM SPSS Inc, Chicago, IL, USA) containing the R package ‘cmprsk’ by Bob Gray. A two-sided *p*-value < 0.05 was considered significant.

## 3. Results

### 3.1. Patient’s Characteristics and Nutritional Assessment

Table 1 shows baseline demographic, clinical, and analytical data for the total studied population, and for patients that had died (*n* = 48; 31.7%) or survived at the end of the five-year follow-up. Patients had a mean age of 69 ± 11 years and were predominantly males, in New York Heart Association (NYHA) functional class II. All patients were treated according to the contemporary guidelines. Abnormal nutritional status was identified in 30 patients (19.8%) with the GLIM criteria in the post hoc analysis, and in 38 patients (25.1%) with the MNA administered at study entry. Accordance between the two methods in the diagnosis of abnormal nutritional status occurred in 11.2% of patients, with an agreement of 77.4% and a kappa index of 0.357 (*p* < 0.001) (Figure 1).

### 3.2. Nutritional Assessment and Prognosis

During the five-year follow-up, 48 patients died (31.7%) and 83 patients were hospitalized, including 39 (25.8%) that were admitted due to HF. The secondary endpoint of cardiovascular mortality occurred in 27 patients (17.8%). In the univariate analysis, we found that abnormal nutrition status identified either by the GLIM criteria or by the MNA was associated with all-cause mortality (GLIM: HR 1.93 (95%CI 1.03–3.60), *p* = 0.038; MNA: HR 2.31 (95%CI 1.28–4.15), *p* = 0.005) (Table 2).

Figure 2 shows survival curves, according to the three categories based on the GLIM criteria (Figure 2A) and MNA merged categories (Figure 2B) of nutritional status. However, in the multivariate analyses, which included age, sex, NYHA functional class, diabetes, and Barthel index, only malnutrition assessed with the MNA remained significant in the model for the primary endpoint of all-cause mortality (Table 2). Moreover, cardiovascular mortality was only related to nutritional status assessed with the MNA but not with the GLIM criteria, both in the univariate and multivariate analyses (Appendix A).

Patients with abnormal nutritional status based on the MNA suffered a twofold higher crude number of recurrent HF-related hospitalizations (17.4 vs. 8.8 per 100 patient-years; *p* = 0.002). In contrast, although patients identified as malnourished with the GLIM criteria had more HF-related hospitalizations than patients with normal nutritional status, the difference did not reach statistical significance (Figure 3). The binomial negative regression analysis showed an IRR of 14.8 (95%CI 6.6–33.1; *p* < 0.001) for MNA and an IRR of 2.16 (95%CI 0.97–4.83; *p* = 0.06) for GLIM. Statistical significance was maintained in the multivariate analysis only for MNA, with an IRR of 10.1 (95%CI 4.5–22.8; *p* < 0.001).

## 4. Discussion

In the present study, we found that nutritional status assessed with the MNA, but not with the GLIM criteria, was an independent predictor of all-cause mortality, CV mortality, and recurrent HF-related hospitalizations over a five-year follow-up of patients with HF that attended our outpatient clinic. To our knowledge, this is the first study that evaluates the GLIM criteria in relation to prognostic significance in outpatients with HF, as well as to compare the GLIM criteria to a previous validated method, such as the MNA, in these patients.

In a recent study, Kootaka and cols. found that malnutrition defined according to the GLIM criteria was a predictor of low physical function and mortality in hospitalized patients with cardiovascular disease [8]. Nevertheless, in contrast to the present study, only 24% of the subjects had prior chronic heart failure, all of them were evaluated during hospital stay, the mean follow-up was relatively short (2 years), and the multivariate analysis was only adjusted for age and sex. Hospitalized patients usually present with acute disease-related malnutrition (DRM) with higher inflammation than chronic DRM observed in ambulatory patients. Furthermore, dietary intake is often reduced in hospitalized patients compared to outpatients due to many reasons, such as periods of fasting in relation to examinations or operations, intense anorexia due to the acute disease, among others [19]. These differences might explain some discrepancies between studies, and they suggest that the GLIM criteria may be more accurate in identifying malnutrition in hospitalized HF patients than in ambulatory subjects with HF. In this regard, the two etiologic components of the GLIM criteria, which are reduced food intake or assimilation and inflammation, could be more easily met by HF inpatients than those in an outpatient setting. On the other hand, phenotypic components of the GLIM criteria are based on BMI and unintentional weight loss. BMI is not a good parameter to assess nutritional status, nor a good parameter to evaluate body composition in subjects with HF, because volume overload can lead to a falsely elevated BMI. This fact, together with the difficulty of assessing non-edematous weight loss in patients with HF, could have influenced our results when evaluating phenotypic components. Consequently, although GLIM-defined malnutrition has been reported to be a predictor of mortality in patients with other chronic diseases, such as cancer [20], it may not perform as well in subjects with HF, probably due to the individual components contained in the GLIM criteria, especially those referring to BMI and weight loss.

In contrast to GLIM criteria, in the present study, the MNA was an independent predictor of all-cause mortality, CV-related mortality, and recurrent HF-related hospitalizations in our HF outpatients during the five-year follow-up. Our results are consistent with those of Bonilla-Palomas et al., who also observed that malnutrition assessed with the MNA was a good predictor of all-cause mortality among patients that attended an HF outpatient clinic [21]. Moreover, in a previous study, we found that MNA was a better predictor of mortality than Subjective Global Assessment, and also had implications on quality of life and physical disability [9].

The prevalence rates of abnormal nutritional status in our cohort differed according to the criteria used (GLIM 19.9% vs. MNA 25.1%). Nevertheless, when the severe malnutrition category was considered, the GLIM criteria identified a higher proportion of patients (5.3%) than MNA (1.3%). Based on malnutrition prevalence alone, and compared to the MNA, the GLIM criteria appeared to under-represent overall malnutrition, but they were more likely to identify severely malnourished individuals. It should be noted that the agreement between GLIM and MNA criteria in our cohort was relatively weak (77.4%). One explanation for this finding could be the method used to identify disease-related inflammation as an etiologic criterion to diagnose malnutrition by GLIM. In our cohort, since all patients had a chronic disease burden, we chose to use CRP as a supportive laboratory marker of inflammation, as per GLIM guidance [7], in order to be more specific. Other authors may exclusively use the underlying chronic disease as an etiological criterion, without taking into account CRP concentrations. The method for evaluating inflammation in our study might have contributed to the underestimation of the prevalence of DRM diagnosed by GLIM criteria in our cohort. Recently, Allard and cols. found a fair sensitivity of the GLIM criteria for diagnosing malnutrition when using Subjective Global Assessment as a comparator in a multicenter prospective study assessing malnutrition at admission [22]. Those authors also used CRP as a supportive measure for inflammation, as in the present study. It would be relevant to further assess the GLIM validity while using the presence of these chronic conditions as a variable.

For all the reasons mentioned above, the MNA might be the best method for assessing nutritional status in HF outpatients, since its results have implications on the prognosis of these patients. Moreover, the MNA questionnaire provides valuable information about the patient’s dietary habits and data that can guide the nutritional treatment approach. Therefore, in our opinion, the MNA should be considered the gold standard to identify malnutrition in the HF population.

Malnutrition status also correlates with an increased risk of death during hospitalization [23,24]. One recent multicenter, randomized, controlled trial in hospitalized HF patients demonstrated that intensive nutritional intervention in malnourished subjects identified by MNA, which included dietary optimization and treatment with oral nutritional supplements if deemed necessary, reduced the risk of all-cause mortality and the readmission rate due to HF worsening when compared to patients receiving conventional management [25]. Given the dire consequences of malnutrition in patients with HF, and the possibility of reversing it with nutritional support, we consider that a nutritional risk screening and, if necessary, a subsequent nutritional assessment with MNA should be included in the comprehensive assessment of patients with chronic HF.

There were several limitations in our study. First, nutritional evaluations were performed only once, at the time of admission; therefore, we had no data on changes in nutritional status during the study period. Second, we used AMC and hand grip strength to assess muscle mass, which might not have been as accurate as other body composition techniques (e.g., computed tomography or dual-energy X-ray absorptiometry). Third, this was a retrospective analysis of data collected during a prospective cohort study, although we performed an extended two to five-year follow-up. Consequently, our results do not allow us to draw definitive conclusions. Finally, the most recent HF therapies shown to be effective in HF patients (i.e., ARNI and iSGLT2) were not used in our study, and we cannot discard that, with the current treatments, results on prognosis could have been different.

## 5. Conclusions

This study showed that malnutrition assessed with the MNA, but not with the GLIM criteria, was an independent predictor of all-cause mortality, cardiovascular-related mortality, and recurrent hospitalizations in outpatients with HF.

The clinical implications of these findings are that a nutritional assessment with the MNA could be essential for the evaluation of ambulatory patients with HF, and that this study allows MNA to be positioned as the preferred method for daily use in these patients.

Future research is needed to investigate the links between nutritional interventions and improvements in survival and HF hospitalizations in outpatients with HF.

## Figures and Tables

**Figure 1 nutrients-14-02244-f001:**
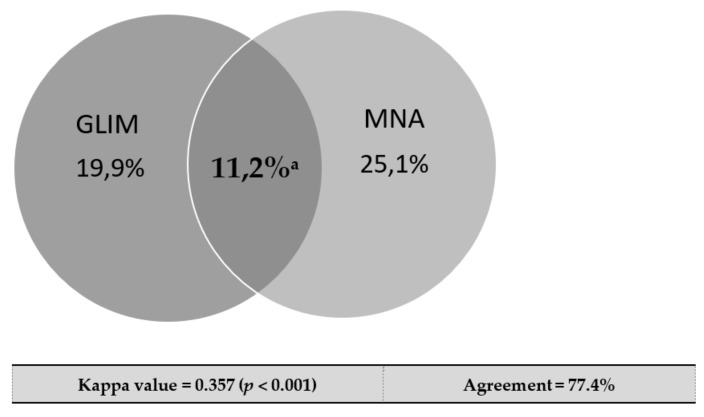
Prevalence of malnutrition in HF outpatients in accordance with GLIM criteria and MNA, and concordance between them. ^a^ Percentage of patients that were identified as malnourished or at nutritional risk by the 2 methods.

**Figure 2 nutrients-14-02244-f002:**
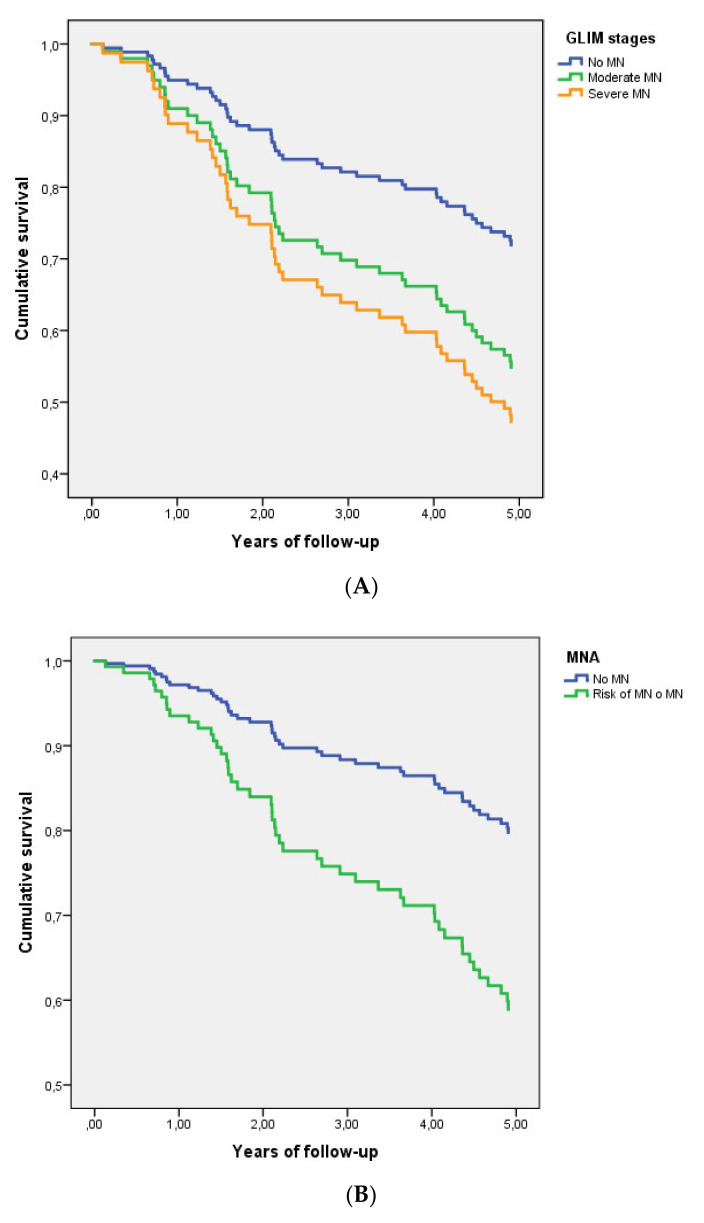
Survival curves (**A**) according to the three categories of nutritional status by the GLIM criteria and (**B**) MNA merged categories of nutritional status (normal nutrition and abnormal nutritional status) in outpatients with HF.

**Figure 3 nutrients-14-02244-f003:**
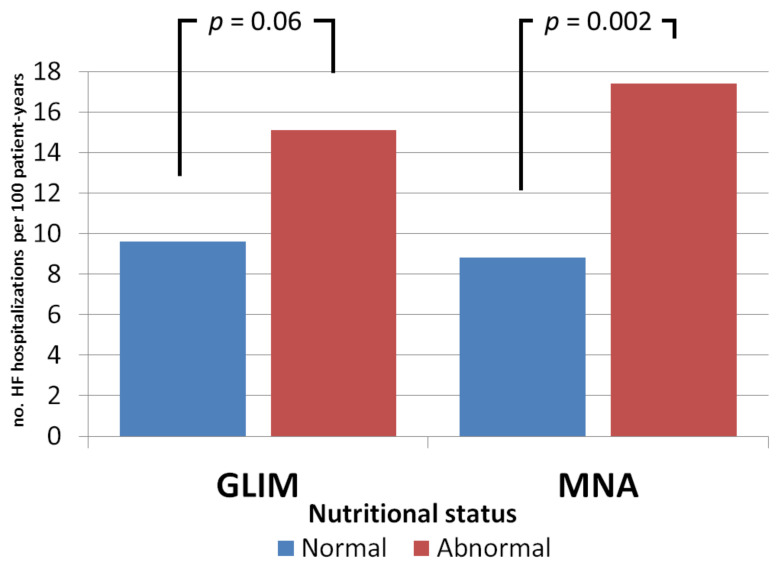
Crude incidence (per 100 patient-years) of recurrent heart failure-related hospitalizations relative to the nutritional status.

**Table 1 nutrients-14-02244-t001:** Demographic and clinical characteristics of patients with HF that died or survived during the five-year follow-up period.

Characteristics	Total	Dead	Survivors	*p*-Value
	N = 151	N = 48	N = 103	
Age (y)	68.6 ± 10.9	75.6 ±8.7	65.4 ± 10.4	<0.001
Male (%)	72.2	70.8	72.8	ns
BMI (kg/m^2^)	27.8 ± 5	27.1 ± 4.7	28.2 ± 5.2	ns
NT-pro BNP (pg/mL)	706.5	1830	448.5	<0.001
	(245.2–1832.5)	(823–3500)	(150.7–1097.5)	
Ejection Fraction (%)	43.9 ± 12.9	46.1	42.9	ns
NYHA (%)				ns
I	4	2.1	4.9	
II	83.4	77.1	86.4	
III ^a^	12.6	20.8	8.7	
Duration of disease (years)	7.0 ± 2.5	9.8 ± 0.4	5.8 ± 2.1	ns
Medication (%)				
ACEI or ARB	84.8	68.8	92.2	<0.001
Beta blocker	88.7	87.5	89.3	ns
Statin	68.2	79.2	63.1	0.048
MNA (%)				0.017
Normal	74.8	62.5	80.5	
At risk of MN	23.8	35.4	18.4	
MN	1.3	2	0.9	
GLIM criteria (%)				ns
Normal	80.1	70.8	84.5	
Moderate MN	14.6	20.8	11.7	
Severe MN	5.3	8.3	3.9	
Hand grip strength	28.6 ± 10.5	23.7 ± 9.1	31.0 ± 10.3	
Kg	−0.4	−0.6	−0.38	<0.001
SD	(−1.07–0.07)	(−1.5–0.05)	(−0.93–0.11)	ns
Barthel index	94 ± 14.2	90.3 ± 17.3	96.2 ± 12.2	0.036
Physical disability ^b^ (%)	23.8	41.7	15.6	0.003
Total cholesterol (mg/dL)	169.4 ± 40.9	154.1 ± 37.7	176.4 ± 40.6	0.002
Lymphocyte (count/mL)	1600	1400	1800	ns
	(1300–2100)	(1020–1700)	(1375–2225)	
Serum albumin (g/dL)	42.5 ± 3.0	42.2 ± 2.5	42.6 ± 3.2	ns

Categorical values are expressed as the percentage (%) of patients; continuous values are expressed as the mean ± SD or the median (25th–75th percentiles), as indicated. BMI = body mass index; NT-pro BNP = N-terminal pro B-type natriuretic peptide; NYHA = New York Heart Association; ACEI = angiotensin-converting enzyme inhibitor; ARB = angiotensin II receptor blockers; MNA = Mini Nutritional Assessment; SGA = Subjective Global Assessment; ns = not statistically significant; MN = Malnutrition. ^a^ No patient was in NYHA functional class IV. ^b^ Physical disability = Barthel index < 100.

**Table 2 nutrients-14-02244-t002:** Cox regression analysis results for factors potentially related to the 5-year all-cause mortality in 151 subjects that attended an outpatient HF clinic.

Factor	5-Year All-Cause Mortality					
	Univariate		Multivariate GLIM		Multivariate MNA	
	HR (95%CI)	*p*-Value	HR 95%CI	*p*-Value	HR 95%CI	*p*-Value
Age	**1.09** **(1.05–1.12)**	**<0.001**	**1.09** **(1.05–1.12)**	**<0.001**	**1.09** **(1.06–1.13)**	**<0.001**
Sex	1.12 (0.59–2.07)	0.73	-	-	-	-
NYHA class	**2.27** **(1.18–4.36**)	**0.013**	**-**	**-**	-	-
MNA ^a^	**2.3** **(1.28–4.15)**	**0.005**	NE	NE	**2.3** **1.23–4.43**	**0.009**
GLIM criteria	**1.93** **(1.03–3.6)**	**0.038**	-	-	NE	NE
Barthel index	**0.98** **(0.96–0.99)**	**0.014**	**-**	**-**	-	-
Diabetes Mellitus	1.73 (0.98–3.05)	0.056	**-**	**-**	-	-

Bold values indicate factors significantly related to mortality. ^a^ For analyses, MNA categories were merged into normal nutritional status and abnormal nutritional status (MN or risk of MN). NYHA = New York Heart Association; MNA = Mini Nutritional Assessment; MN = Malnutrition; *p*-values are based on Cox regression analysis. NE = non-evaluated.

## Data Availability

The data presented in this study are available on request from the corresponding author.

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
