# Peer review of "Nutritional Status According to the GLIM Criteria in Patients with Chronic Heart Failure: Association with Prognosis"

_nutrients, 2022, doi:10.3390/nu14112244_

Round 1

Reviewer 1 Report

Dear authors, I had the privilege to review your manuscript entitled “Nutritional status according to the GLIM criteria in patients with chronic heart failure: association with prognosis”. In this post-hoc analysis conducted on a small group of HF ambulatory patients enrolled in 2013 (n=131), you demonstrated how malnutritional status assessed by MNA carried a stronger prognostic role than malnutritional status assessed by the new standard of care (GLIM).

The low overlap between the 2 diagnosis (just 77%) support the different diagnostic accuracy of malnutritional status according to the definition used.

The article is well written and clear in its passages. The main message is clear.

Some point must be address:

  • The patients were enrolled in 2013. HF therapies and HF prognosis have changed dramatically in the last 5-10 years. Why don’t you include more patients in the analysis? At least until 2018
  • Moreover, since you have approximately 8 years follow-up I suggest to update your follow-up data up to 2022;
  • Was a sample calculation done? Are 131 patients enough to support your conclusions? This should be done and commented in the methods section. Otherwise, this study can be just a hypothesis generating study
  • Few sentences are repeated in the Introduction and in the discussion section. Please fix
  • Given the high mortality rate the “malnutrition status” looks more like cardiac cachexia, a condition related to end-stage heart failure. This condition is a prognostic marker per se and, from a pathophysiological point of view, is more related to a severe cardiac insufficiency more than low calories intake. This should be adequately commented also because cardiac cachexia can be hardly corrected with therapeutic intervention, including rehabilitation

Reviewer 2 Report

Introduction

I suggest to mention that the malnutrition status also correlates with an increased risk of death during hospitalisation.

A good references are:

Li H, Zhou P, Zhao Y, Ni H, Luo X, Li J. Prediction of all-cause mortality with malnutrition assessed by controlling nutritional status score in patients with heart failure: a systematic review and meta-analysis. Public Health Nutr. 2021 Jun 30:1-8. doi: 10.1017/S1368980021002470.

Czapla M, Juárez-Vela R, Łokieć K, Karniej P. The Association between Nutritional Status and In-Hospital Mortality among Patients with Heart Failure-A Result of the Retrospective Nutritional Status Heart Study 2 (NSHS2). Nutrients. 2021 May 14;13(5):1669. doi: 10.3390/nu13051669.

Kałużna-Oleksy, M.; Krysztofiak, H.; Migaj, J.; Wleklik, M.; Dudek, M.; Uchmanowicz, I.; Lesiak, M.; Straburzyńska-Migaj, E. Relationship between Nutritional Status and Clinical and Biochemical Parameters in Hospitalized Patients with Heart Failure with Reduced Ejection Fraction, with 1-year Follow-Up. Nutrients 202012, 2330. https://doi.org/10.3390/nu12082330

Methodology:

The description and application of the measurement methods are appropriate but I think that you should add in methodology information that MNA form provided by Société des Produits Nestlé SA 1994, Revision 2009, Vevy, Switzerland, Trademark Owners, which holds the copyright of the instrument: http://www.mna-elderly.com/) – in this website describes all rules how to citate this form.

Result: Generally the analysis of the results was carried out in detail. In tabl1 In my opinion is better to use words “Survivors” instead “alive”

Discussion

The discussion was conducted in an appropriate manner.

The work is limited by no data on changes in nutritional status during the study period but all limitation also noted by the authors.

Generally The final conclusions are correct but I recommended create separate part for clinical implication and conclusion.

Round 2

Reviewer 1 Report

Dear authors, the articles has been improved following the comments. I think it can be accepted for publication in Children.